# Speech Watermarking Method Using McAdams Coefficient Based on Random Forest Learning

**DOI:** 10.3390/e23101246

**Published:** 2021-09-25

**Authors:** Candy Olivia Mawalim, Masashi Unoki

**Affiliations:** School of Information Science, Japan Advanced Institute of Science and Technology, Ishikawa 923-1292, Japan; unoki@jaist.ac.jp

**Keywords:** speech watermarking, McAdams coefficient, random forest classifier, machine learning for watermarking

## Abstract

Speech watermarking has become a promising solution for protecting the security of speech communication systems. We propose a speech watermarking method that uses the McAdams coefficient, which is commonly used for frequency harmonics adjustment. The embedding process was conducted, using bit-inverse shifting. We also developed a random forest classifier, using features related to frequency harmonics for blind detection. An objective evaluation was conducted to analyze the performance of our method in terms of the inaudibility and robustness requirements. The results indicate that our method satisfies the speech watermarking requirements with a 16 bps payload under normal conditions and numerous non-malicious signal processing operations, e.g., conversion to Ogg or MP4 format.

## 1. Introduction

Speech communication technology has greatly advanced, due to its application in daily life. This technology is usually implemented via a communication channel, such as the public switched telephone network (PSTN) and voice over internet protocol (VoIP). The speech communication channel is considerably vulnerable to attacks; thus, protection and prevention countermeasures are indispensable in speech research. For instance, the recent technologies for voice conversion and text-to-speech systems are capable of using speech tampering or spoofing [1,2].

Methods for promoting secure speech communication systems are classified into two main categories, i.e., cryptography and information hiding. Cryptography-based methods convert speech data to another form that can be accessed only by an authorized person with a private key. These methods are useful for specific applications that can afford the additional computational time and complexity of the cryptography process. However, protection is limited only once the content is in an encrypted state [3]. Information-hiding–based methods preserve the privacy and security of speech data by imperceptibly embedding particular information that needs to be hid [4,5]. Two information-hiding categories are steganography and watermarking, depending upon the purpose.

Information-hiding–based methods for speech signals were developed within the past approximately 25 years [4]. Speech information hiding should satisfy at least three requirements: inaudibility (manipulation does not cause distortion perceivable by the human auditory system), blindness (accurate detection without the original signal), and robustness against common signal processing operations. Well-known classical speech information hiding methods are least significant bit (LSB), phase modulation, and direct spread spectrum (DSS) [4]. Although each method has advantages, they have shortcomings and need improvement, especially in balancing the trade-off between inaudibility and robustness. To compensate for the shortcomings of these methods, psychoacoustics are often considered [4,5,6]. For instance, speech watermarking based on cochlear delay was proposed to improve the phase modulation method [6].

Another approach to improve the trade-off between inaudibility and robustness is considering the features used in speech codecs [2,7,8]. Speech codecs are widely applied before speech is transmitted through a communication channel. Thus, using features in speech codecs for speech information hiding improves robustness. Line spectral frequencies (LSFs) are used as features in speech codecs with several speech watermarking methods [8,9,10]. LSFs can be directly modified in accordance with a particular speech codec quantization method [8] or manipulated accordingly to control speech formants for representing hidden information [9,10].

We investigate a parameter that affects the formation of auditory images, namely, the McAdams coefficient [11], for speech watermarking in this study. The modification of the McAdams coefficient is useful for adjusting frequency harmonics in the audio signal [12,13]. It was also introduced for de-identifying or anonymizing speech signals [3,14,15]. Since the McAdams coefficient is related to the adjustment of frequency harmonics (related to LSFs), we hypothesize that this coefficient is suitable for speech watermarking.

Another novelty present in this study is that we propose a speech watermarking method based on a machine learning model. Studies on digital image watermarking based on machine learning models have shown impressive results [16,17]. However, due to the higher complexity of speech compared to image data, machine learning models for speech watermarking have not been widely explored [18]. We constructed a machine-learning–based blind detection model by using a binary classification task based on a random forest algorithm (hereafter, we refer to this model as random forest classifier). We hypothesize that this classifier can automatically generate the rules for blind detection by using the acoustic cues related to the McAdams coefficient.

The remainder of this paper is organized as follows. In Section 2, we review related work on speech processing, using the McAdams coefficient. In Section 3, we present our proposed speech watermarking method. In Section 4, we describe the settings for our experiments on evaluating the proposed method; we present the results, application, and remaining limitations in Section 5. Finally, we conclude our paper in Section 6.

## 2. Speech Processing Using McAdams Coefficient

In 1984, Stephen McAdams investigated the acoustic cues that affect the formation of auditory images [11]. These acoustic cues are comprised of frequency harmonicity, coherence of low-frequency modulation, and stability of spectral form when combined with frequency modulation. He carried out several listening evaluations and reported the characteristics of these acoustic cues. His findings contributed significantly to the speech signal processing field, e.g., music signal processing [12] and speaker de-identification [13,14].

One of the techniques of generating sounds is based on additive synthesis [12] and is commonly used in music signal processing for timbre generation through the combination of multiple consinusoidal oscillations by an inverse Fourier series with magnitude and phase shift. Mathematically, the additive synthesis process is expressed as follows:(1)$$y_{syn}\left(t\right)=\sum_{h=1}^{H}r_{h}\left(t\right)\cos\left(2\pi {\left(hf_{0}\right)}^{\alpha }t+\Phi_{h}\right),$$
where $y_{syn}\left(t\right)$ is the synthesized signal, *h* is the harmonic index, $r_{h}\left(t\right)$ is the amplitude, $\Phi_{h}$ is the phase, and α is the McAdams coefficient [11]. The McAdams coefficient is used for adjusting frequency harmonics to non-harmonics components that affect the perception of timbre.

In a study on speaker de-identification or speaker anonymization, a similar technique for frequency harmonics adjustment was considered for vocal-tract-length normalization [13]. Subsequently, a study on speaker anonymization using the McAdams coefficient was carried out [14]. The McAdams coefficient was manipulated to alter the formant position of the original speech at the frame level on the basis of linear predictive coding (LPC) analysis and a synthesis technique. Figure 1 shows the procedure of McAdams coefficient modification for speaker anonymization [14]. In speaker anonymization, the McAdams coefficient is manipulated as far as possible from the original speech (α=1) with consideration of the sound distortion caused. For example, the anonymization introduced as the secondary baseline in the Voice Privacy Challenge 2020 [19] used fixed α=0.8 [3]. Those results indicate the degree of McAdams coefficient manipulation on our perception of speaker individuality [14].

## 3. Proposed Method

We developed our speech watermarking method through McAdams coefficient manipulation. In previous studies, the McAdams coefficient was modified to preserve speaker identity (speaker anonymization) [3,14,15]. The further away the McAdams coefficient is from the original speech (α=1), the greater the level of anonymization (better performance in reducing speaker verifiability metrics) [14]. However, this advantage results in more speech distortion (degrades speech intelligibility and naturalness). Too much distortion is non-compensable for speech watermarking since speech quality relates to one of the most important requirements in speech watermarking, i.e., inaudibility. Therefore, careful consideration should be made for speech watermarking based on McAdams coefficient modification.

We previously proposed a watermarking framework for improving the security of McAdams-coefficient–based speaker anonymization [15]. Two fixed McAdams coefficients were used to represent the binary bit information for speech watermarking. These values were chosen on the basis of the second baseline speaker anonymization system in the Voice Privacy Challenge 2020 [3] and the optimal gap for stochastic McAdams-coefficient–based speaker watermarking [14]. The watermarking processes were conducted in a manner similar to signal modulation. The watermark detection process was based on the threshold of a certain parameter. The experimental results indicated that our watermarking framework could be applied to improve the security of speaker anonymization with a limitation of relatively low payload.

In contrast to the related studies on speaker anonymization [3,14,15], we considered a McAdams coefficient closer to the original speech and a smaller shift to maintain the inaudibility criteria on speech watermarking. A smaller shift means that the McAdams coefficient for representing bit-0 ($\alpha_{0}$) is close to that representing bit-1 ($\alpha_{1}$). We developed a random forest classifier to detect embedded watermarks. We then investigated the optimal McAdams coefficient that can balance inaudibility with the blind-detectability robustness. This section is comprised of three sub-sections, i.e., manipulation of the McAdams coefficient, the data-embedding process, and data-detection process.

### 3.1. McAdams Coefficient Manipulation

The manipulation of the McAdams coefficient follows the block diagram shown in Figure 1. The original signal in the time domain (x(n)) is first divided into several overlap frames. Each speech frame is then passed through a linear prediction (LP) analysis filter, which is an all-pole filter that mimics the source-filter analysis model of a speech production system. In this study, we used the LP order of 12 (M=12). The LPC analysis is characterized by the following differential equation:(2)$$s\left(n\right)=\sum_{i=1}^{M}c\left(i\right)s\left(n-i\right)+e\left(n\right),$$
where s(n) is the speech frame, c(i) is the *i*-th order LP coefficient, *M* is the order of LP, and e(n) is the prediction error (residuals). The corresponding transfer function (H(z)) for Equation (Equation 2) is represented using all-pole autoregressive filters as follows:(3)$$H\left(z\right)=\frac{1}{1-\sum_{i=1}^{M}c\left(i\right)z^{-i}}.$$

The LP coefficients (c(i)) obtained from the LPC analysis are used to derive the poles (ϕ). The derived poles can be categorized into complex and real poles. Complex poles have non-zero imaginary values, whereas real poles have a zero-valued imaginary term. The McAdams coefficient (α) corresponds to the power of complex poles. The manipulation of alpha results in angle shifting of complex pole positions ($\phi^{\alpha }$) either clockwise or counter-clockwise, depending on α and ϕ [14]. When α<1, $\phi^{\alpha }$ is in the counter-clockwise direction when ϕ<1 radian and in the clockwise direction when ϕ>1 radian. The opposite direction applies when α>1. We investigate McAdams coefficient manipulation when α<1 in this study. Figure 2 shows the poles location and frequency-response envelopes obtained from original signal and McAdams coefficient manipulation when α={0.85,0.9,0.95}.

After shifting complex pole locations by manipulating the McAdams coefficient, both complex and real poles are converted to new LP coefficients ($c^{\prime }\left(i\right)$). These LP coefficients and the original residuals (e(n)) are resynthesized as modified speech frames. Finally, the modified speech frames are concatenated, using the overlap and add technique to generate the modified speech signal (a(n)).

### 3.2. Data-Embedding Process

Figure 3 the block diagram of the data-embedding process. This embedding process is based on signal modulation (similar to our previous study [15]). Generally, we use two McAdams coefficients to represent binary information (bit “0” and bit “1”). The watermarked bit-stream (w(k)), which is comprised of binary information, is embedded in the following steps:

**Step 1:** The original signal (x(n)) is modified on the basis of the McAdams coefficient manipulation process explained in Section 3.1. Two McAdams coefficients are used in the embedding process to represent binary bit-“0” ($\alpha_{0}$) and bit-“1” ($\alpha_{1}$). The gap between $\alpha_{0}$ and $\alpha_{1}$ can be regarded as the scaling factor of watermarking. A larger gap creates a stronger watermark but increases distortion. Next, a normalization method is applied based on the ratio of the power spectral density of both modified signals to compensate for the gap in spectral shift at frame transition. The results of this step are two speech signals with new McAdams coefficients, ($a_{0}\left(n\right)$ and $a_{1}\left(n\right)$).

**Step 2:** The watermarked bit-stream (w(k)) is set in accordance with the hidden information in a binary stream representation, e.g., w(k)={1,0,0,1,0,1,1,1,0,1,1,0,1}.

**Step 3:** The watermarked signal (y(n)) is determined by bit-inverse switching between the modified signals ($a_{0}$ and $a_{1}$) and the watermarked bit-stream (w(k)). For example, if the current bit is “1”, the current speech frame is set to the speech frame derived using $\alpha_{1}$. We concatenate all the speech frames obtained from this process to be y(n).

### 3.3. Data-Detection Process

As shown in Figure 2, McAdams coefficient manipulation causes the shifting in pole locations and frequency-response envelopes. We thus investigated the statistical properties of these cues for the data-detection process. In contrast to the common watermark detection methods that are based on a threshold or a fixed set of rules, a machine learning model is constructed to blindly detect watermarks on the basis of those cues as features and is based on a random forest algorithm [20] (as shown in Figure 4). The random forest classifier generates a number of decision tree classifiers on random sub-samples of the training dataset to control overfitting and improve the prediction accuracy.

Before deciding on a random forest algorithm for generating our data-detection model, we carried out a preliminary experiment. We compared three watermark detection methods: (1) using a rule-based model with thresholds on power spectral density and pole location; (2) using a decision tree model; and (3) using a random forest model. The features for constructing the decision tree and random forest models are power spectral density, pole locations, and statistical features (minimum, maximum, mean, standard deviation, skewness, and kurtosis) of the frequency-response envelope of the watermarked speech frames (without any pre-processing). We evaluated these three methods, using a dataset consisting of 100 utterances randomly selected from the LibriSpeech [21] and VCTK [22] corpora. These 100 utterances were selected from one female speaker (LibriSpeech) and one male speaker (VCTK). We set the watermarking payload to 4 bps, a fixed set of McAdams coefficients for watermarking ($\left\{\alpha_{0},\alpha_{1}\right\}=\left\{0.9,1\right\}$), and a fixed frame size (20 ms) without a sliding window. The average classification errors of methods, using rule-based, decision tree model, and random forest models in a 10-fold cross-validation evaluation were 32.12%, 26.25%, and 16.42%, respectively. On the basis of these results, we chose the random forest algorithm because it is the most stable and robust against outliers, compared to the others used in our preliminary experiment.

To improve our random forest classifier model for the blind-detection process, we use a short-term analysis frame with a fixed length (default frame size = 20 ms) with a sliding window. The features for constructing this data-detection model are power spectral density, complex pole locations, and statistical features of line spectral frequencies (LSFs) pairs on the frequency-response envelope. The statistical features consist of mean, standard deviation, skewness, and kurtosis. The statistical features of LSFs are used because they are more relevant than the global statistical features of the frequency-response envelope to represent the McAdams coefficient manipulation. Figure 5 shows the LSF positions on the frequency-response envelope of modified speech signals when α={1,0.95,0.9,0.85}. We generate two modified speech signals through McAdams coefficient modification ($a_{0}\left(n\right)$ and $a_{1}\left(n\right)$), following the process explained in Section 3.1 for the training process. The label corresponds to the binary bit represented by the McAdams coefficient (“0” or “1”).

Figure 6 shows the block diagram of the data-detection process. The details of this process are as follows:

**Step 1:** The watermarked signal (y(n)) is passed through a pre-emphasis filter. This filter is used to compensate for the average spectral shape that emphasizes the higher frequency components. A finite impulse response (FIR) filter is used as the pre-emphasis filter (P(z)), which is expressed as follows:(4)$$P\left(z\right)=1-0.95z^{-1}.$$

**Step 2:** Since the constructed random forest classifier works on a short-time frame basis, we use the sliding window technique to obtain more analysis speech frames for optimizing the data-detection process. For example, if the sampling frequency (Fs) is 16 kHz and default frame size is set to 20 ms, we have almost double the number of speech frames when the shift length is set to half the frame size. Figure 7 illustrates the data-detection process using a sliding window.

**Step 3:** From each watermarked speech frame obtained from the sliding window, we conduct feature extraction, i.e., complex pole positions, statistical features of LSF pairs from the frequency-response envelope, and power spectral density. The complex pole positions and statistical features of LSF pairs from the frequency-response envelope are derived from LP analysis. The power spectral density is obtained, using the Fourier transform.

**Step 4:** Finally, our random forest classifier is used to generate the detected watermarked bit-stream ($w^{\prime }\left(k\right)$) on the basis of the majority voting of the detected bit in all sliding window sub-frames in the corresponding watermarked frame. The most common bit information shown in Figure 7 in five detected bits from the classification task of five sub-frames determine the detected watermark bit of the first frame.

## 4. Experimental Setup

This section describes the dataset, random forest classifier for the data-detection process and evaluation setting to analyze the performance of our proposed method.

### 4.1. Dataset

We semi-randomly selected 250 utterances from LibriSpeech [21], and 250 utterances from VCTK [22]. Semi-randomly means that we selected utterances from a particular number of speakers (with balance gender distribution). LibriSpeech is sampled at 16 kHz and designed for automatic speech recognition research, and VCTK is sampled at 48 kHz and designed for text-to-speech research. We unified the sampling rate of both corpora to 16 kHz. The selected utterances varied depending on the speaker and speech content.

Table 1 shows the distribution of the dataset. LibriSpeech has fewer utterances but a relatively long duration, while VCTK has more utterances (almost 10 times that of LibriSpeech) but is relatively shorter in duration. Due to these differences, we used a different number of speakers from each corpus. A total of 500 utterances were then split into 90% for the training set and 10% for the testing set. The training set was used for constructing the random forest classifier for blind detection. The testing set was then used for evaluating speech watermarking performance.

### 4.2. Evaluation Setting

The testing set consisted of 50 utterances. The objective evaluation of our proposed speech watermarking method was based on the information-hiding criteria suggested in [23]. There were two main goals for this evaluation, i.e., (1) to investigate the trade-off between inaudibility and detection rate of our method using various gaps between McAdams coefficients; and (2) investigate the robustness of our method against various speech processing operations.

To reach the first goal, we considered five different McAdams coefficients ($\alpha_{0}$ = {0.95,0.925,0.9,0.875,0.85}) as representations of bit “0”, where we kept $\alpha_{1}=1$ as a representation of bit “1”. These values were chosen to analyze the optimal gap to balance the inaudibility and robustness requirements. We thus constructed five random forest classifiers for blindly detecting the watermarks with regard to the McAdams coefficient. The classification errors of all random forest classifiers are shown in Figure 8. The metrics for evaluating the inaudibility requirement are log spectral distance (LSD) [24] and perceptual evaluation of speech quality (PESQ) [25] ITU-T P.862. LSD is used to measure the spectral distortion of the watermarked signal (y(n)) in comparison with the original signal (x(n)) in decibels (dB), as follows:(5)$$\mathrm{LSD}\left(X,Y\right)=\sqrt{\frac{1}{K}\sum_{k=1}^{K}{\left(10\log_{{}_{10}}\frac{{\left|X\left(\omega ,k\right)\right|}^{2}}{{\left|Y\left(\omega ,k\right)\right|}^{2}}\right)}^{2}},$$
where X(ω,k) and Y(ω,k) are the short-time Fourier transform of the original (x(n)) and modified signal (y(n)) of the *k*-th frame, respectively. The inaudibility threshold for LSD is typically 1 dB. PESQ represents the perceptual speech quality of y(n) with x(n) as the reference in mean opinion scores (MOS). The MOS varies from a scale of 1 (bad) to 5 (excellent). Typically, the PESQ threshold for speech watermarking is 3 (fair or slightly annoying).

For evaluating watermark detection accuracy and security level, we used the bit error rate (BER), false acceptance rate (FAR), false rejection rate (FRR), and F1-score. The threshold set for an acceptable BER is 10% [23]. In the evaluation, we defined the watermarked bit-stream (w(k)) as a random binary stream with the length depending on the payload. We investigated the payloads of 2, 4, 8, 16, and 32 bps. For robustness evaluation, we considered eight cases of non-malicious signal processing operations, i.e., normal (no attack), down-sampling to 12 kHz (resample-12), up-sampling to 24 kHz (resample-24), bit compression to 8 bits (requant-8), bit extension to 24 bits (requant-24), conversion to Ogg format (Ogg), conversion to MPEG-4 Part 14 or MP4 format (MP4), and conversion to G723.1 codec (G723). The bitrate of the G723.1 codec is 5.3 kbps with the algebraic code-excited linear prediction (ACELP) algorithm.

We also carried out a comparison analysis among our proposed method, using the McAdams coefficients $\left(\alpha_{0},\alpha_{1}\right)=\left(0.9,1.0\right)$ (Proposed) and two other well-known speech watermarking methods, i.e., LSB and DSS. These two methods were chosen because they can clearly represent the inaudibility and robustness trade-off. LSB works by modifying the most insignificant bits of the speech signal with watermarks, thus achieving high performance in inaudibility requirements, but it is very fragile against any signal processing operation. In contrast, DSS works by spreading the watermarks over the whole frequency band. Therefore, it is preferred due to its robustness, but it causes significant distortion throughout the speech (lack of inaudibility). We conducted the comparative analysis using payloads of 4, 8, 16, and 32 bps.

## 5. Results

Figure 9 shows the watermark detection accuracy and security level results in terms of BER, FAR, FRR, and F1-score. Five McAdams coefficients were used to represent bit “0” ($\alpha_{0}$ = {0.95,0.925,0.9,0.875,0.85}), whereas the McAdams coefficient for representing bit “1” was set to 1 ($\alpha_{1}=1$). The results indicate a similar tendency for all these metrics when using a larger gap between $\alpha_{0}$ and $\alpha_{1}$, i.e., better detectability, except a slight anomaly in FAR for payloads of 16 and 32 bps. Considering the detectability threshold (BER = 10%), only when $\alpha_{0}=0.85$, the embedding payload was up to 32 bps. With $\alpha_{0}=\left\{0.875,0.9\right\}$, the payload was 16 bps. For other observed $\alpha_{0}$, the payload was less than 16 bps. A similar error rate for FAR and FRR was also found when we considered the observed payloads. When considering the overall security level in F1-score with a threshold of 90%, the proposed methods with $\alpha_{0}\leq 0.9$ reached a payload of 16 bps.

The results of the inaudibility test are shown in Figure 10. On the basis of the inaudibility threshold, the evaluation results indicate that with $\alpha_{0}$ ≤ 0.9, both PESQ and LSD scores satisfied the requirement of up to 32 bps. The inaudibility requirement could be satisfied by watermarked signals with $\alpha_{0}=0.875$ up to 16 bps and $\alpha_{0}=0.85$ up to 8 bps. We will thus consider using $\alpha_{0}=0.9$ for further analysis of robustness. As a reference, we provide demo speech outputs from our proposed method that can be accessed publicly at [26].

The robustness results in eight cases are shown in Figure 11. The watermarked signal was generated with $\alpha_{0}=0.9$ and $\alpha_{1}=1$. By only considering the detectability and security level threshold, the results indicate that our proposed method had similar robustness with the normal case when dealing with up-sampling (resample-24), bit extension (requant-24), Ogg, and MP4 processing operations. Robustness degraded when down-sampling (resample-12), bit compression (requant-8), and G723.1 codec (G723) were applied. For resample-12 and requant-8, we can say that our proposed method is robust when the payload is 4 bps (BER <10%). Unfortunately, our proposed method is not robust when the G723.1 codec is applied (BER >10%).

The results on the comparative analysis among our proposed method with $\left(\alpha_{0},\alpha_{1}\right)=\left(0.9,1.0\right)$ (proposed), LSB, and DSS are shown in Figure 12 and Figure 13. Figure 12 shows the inaudibility comparison results in terms of PESQ and LSD. These results indicate that LSB and our proposed method could pass the threshold of inaudibility but not DSS.

Figure 13 shows the robustness comparison results in terms of BER. In contrast to the inaudibility results, the robustness results indicate that LSB was fragile in dealing with almost all observed signal processing operations, except with the up-sampling to 24 kHz. However, DSS was very robust even in a higher payload, except with the G723.1 speech codec. Although not as robust as DSS, our proposed method had better robustness against most of the observed signal processing operations (down-sampling, re-quantization, Ogg format, and MP4 format) than LSB.

To better represent the application of speech watermarking, we embedded an image as a watermark to a speech signal. The watermark detection results are shown in Figure 14. The size of the image in the binary bit-stream was 80×192. The watermarked signal was generated using $\alpha_{0}=0.9$ and $\alpha_{1}=1$ with 4-bps payload. Although not perfectly accurate, we could observe the reflection of embedded image information, even after certain operations, including re-sampling, re-quantization, and conversion to Ogg and MP4 formats.

## 6. Conclusions

We proposed a speech watermarking method that is based on the McAdams coefficient, using bit-inverse embedding and blind detection by using a random forest classifier. We conducted an evaluation to investigate the trade-off between inaudibility and watermark-detection rate using various gaps in the McAdams coefficients for representing the watermark bit-stream. We also conducted a robustness evaluation by considering several signal processing operations. The results indicate that our method with a McAdams coefficient gap of 0.1 satisfied the detectability and inaudibility requirements up to 16 bps. Our method was also robust against most observed signal processing operations, except for the G723.1 codec. For future work, we will improve the speaker anonymization of our method, which has a limitation of relatively low payload, as determined in our previous study [15].

## Figures and Tables

**Figure 1 entropy-23-01246-f001:**
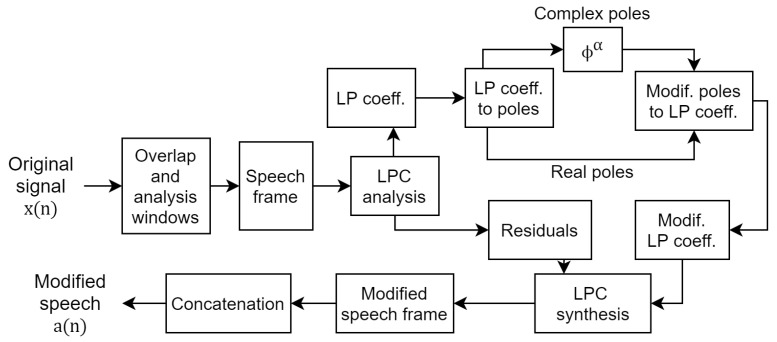
Block diagram of McAdams coefficient modification.

**Figure 2 entropy-23-01246-f002:**
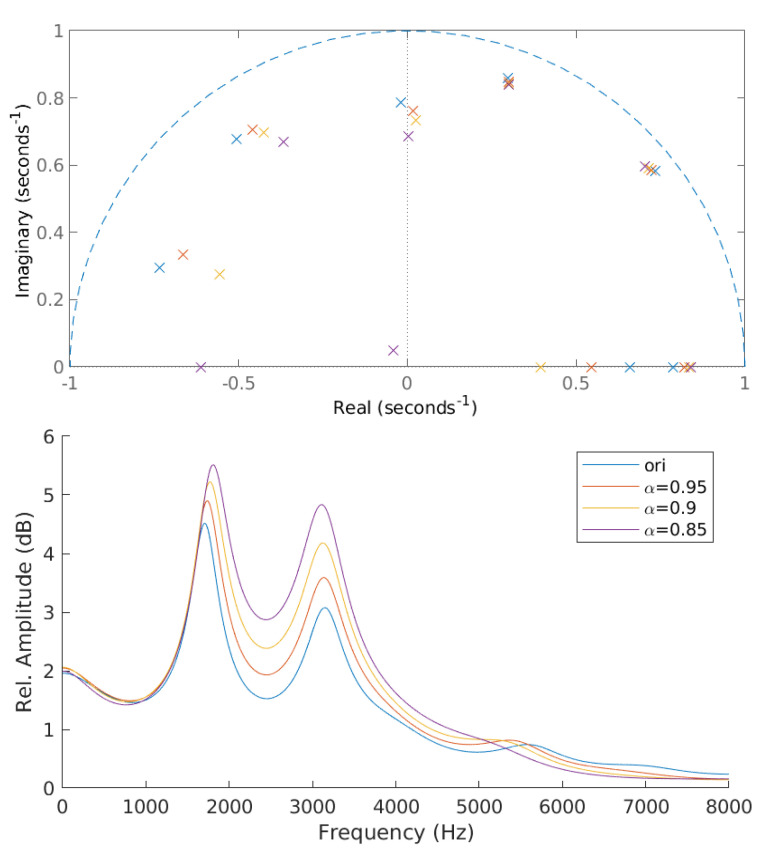
Pole locations and frequency-response envelopes of original signal (ori) and modified signals with McAdams coefficients (α={0.85,0.9,0.95}).

**Figure 3 entropy-23-01246-f003:**
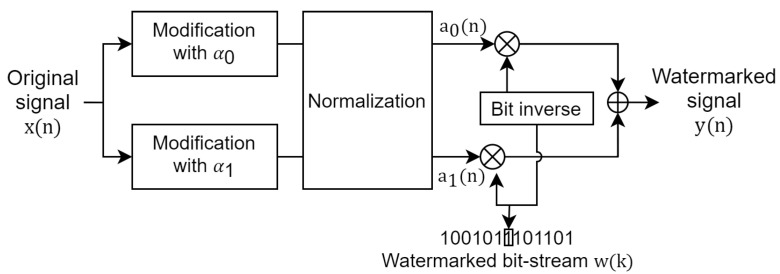
Block diagram of data-embedding process.

**Figure 4 entropy-23-01246-f004:**
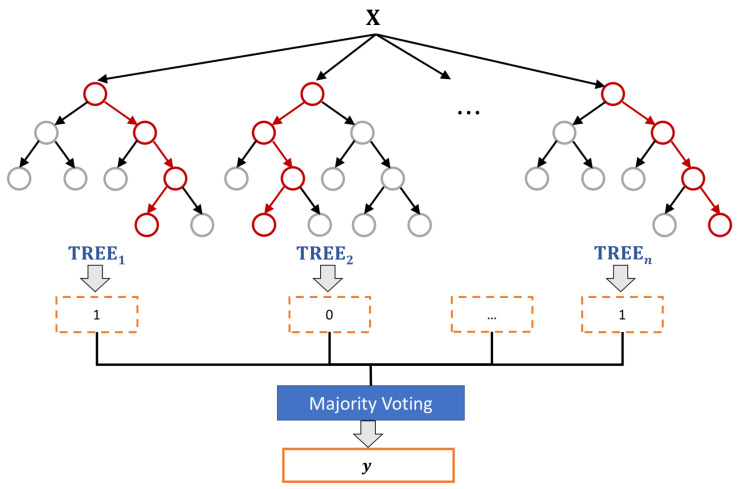
Random forest classifier for data detection. **X** is the set of features, ***y*** is the classification label (“0” or “1”), and *n* is the number of trees.

**Figure 5 entropy-23-01246-f005:**
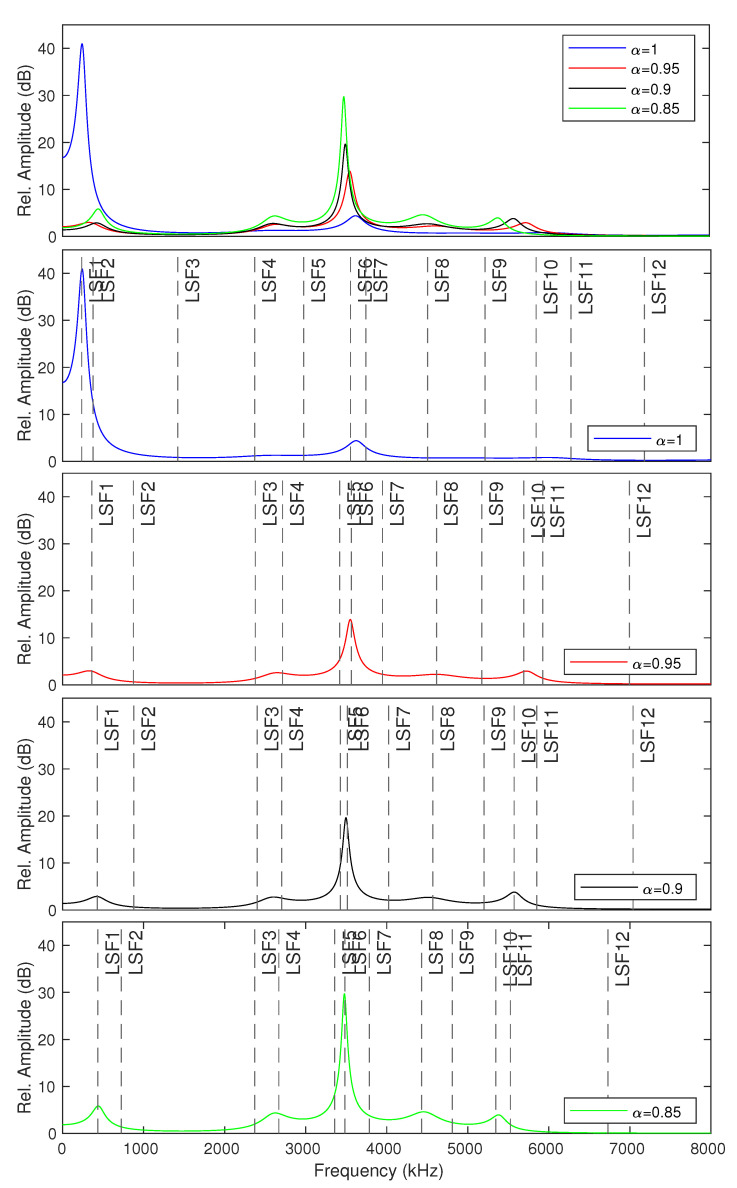
LSF positions on frequency-response envelopes obtained from various McAdams coefficients (α={1,0.95,0.9,0.85}).

**Figure 6 entropy-23-01246-f006:**
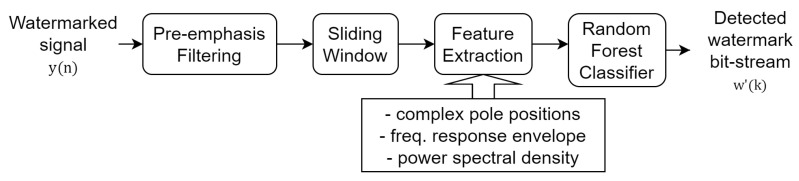
Block diagram of blind-detection process. $w^{\prime }\left(k\right)$ is the detected watermark bit-stream of *k*-th frame.

**Figure 7 entropy-23-01246-f007:**
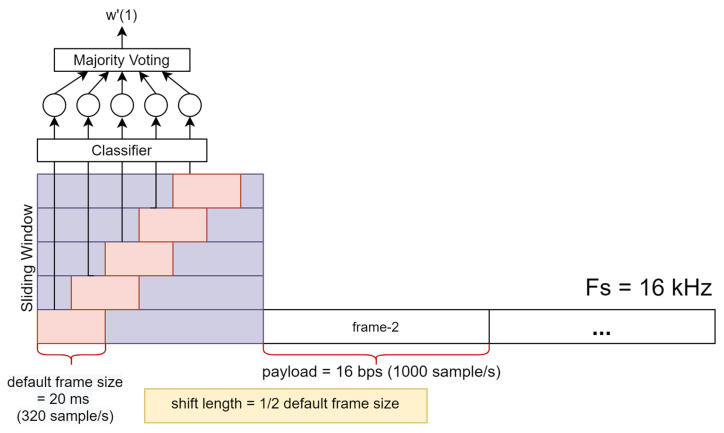
Illustration of watermark detection using sliding window. Sampling frequency (Fs) is 16 kHz, payload is 16 bps, and shift length is set to half the default short-time frame size (10 ms).

**Figure 8 entropy-23-01246-f008:**
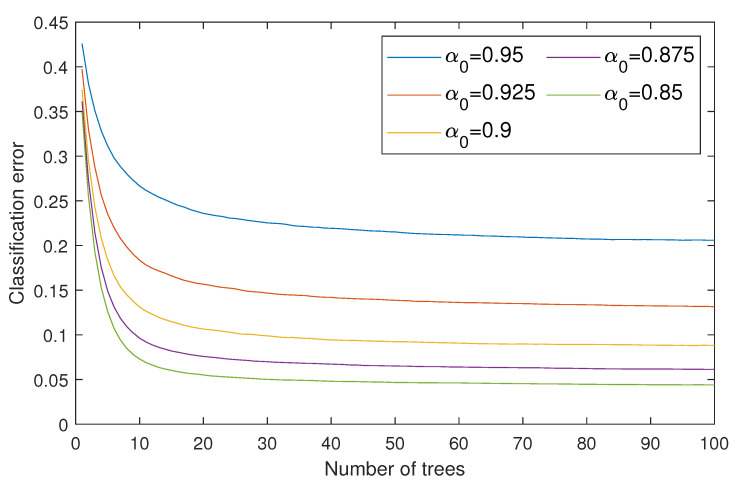
Classification errors of constructed random forest classifiers using several McAdams coefficients for representing bit-“0” ($\alpha_{0}=\left\{0.95,0.925,0.9,0.875,0.85\right\}$). Maximum number of trees was set to 100.

**Figure 9 entropy-23-01246-f009:**
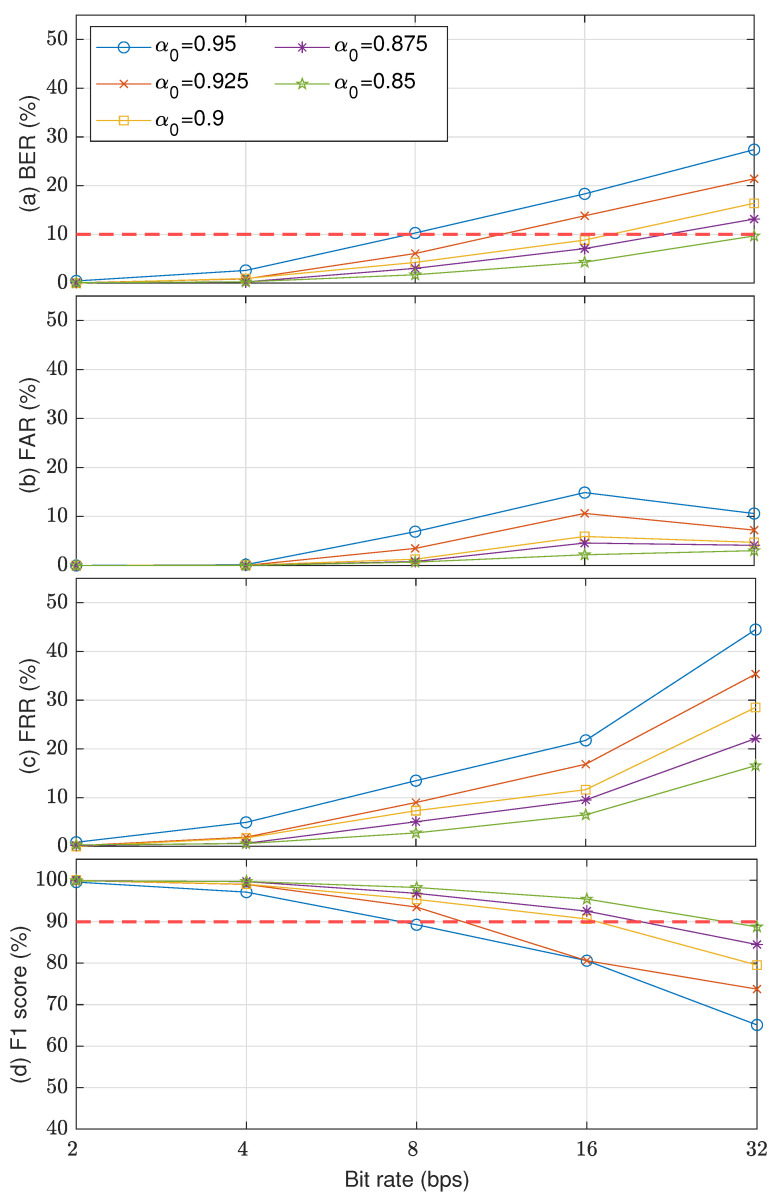
Watermark detection accuracy results using several McAdams coefficients for representing bit “0” ($\alpha_{0}=\left\{0.95,0.925,0.9,0.875,0.85\right\}$) in terms of (**a**) BER, (**b**) FAR, (**c**) FRR, and (**d**) F1-score.

**Figure 10 entropy-23-01246-f010:**
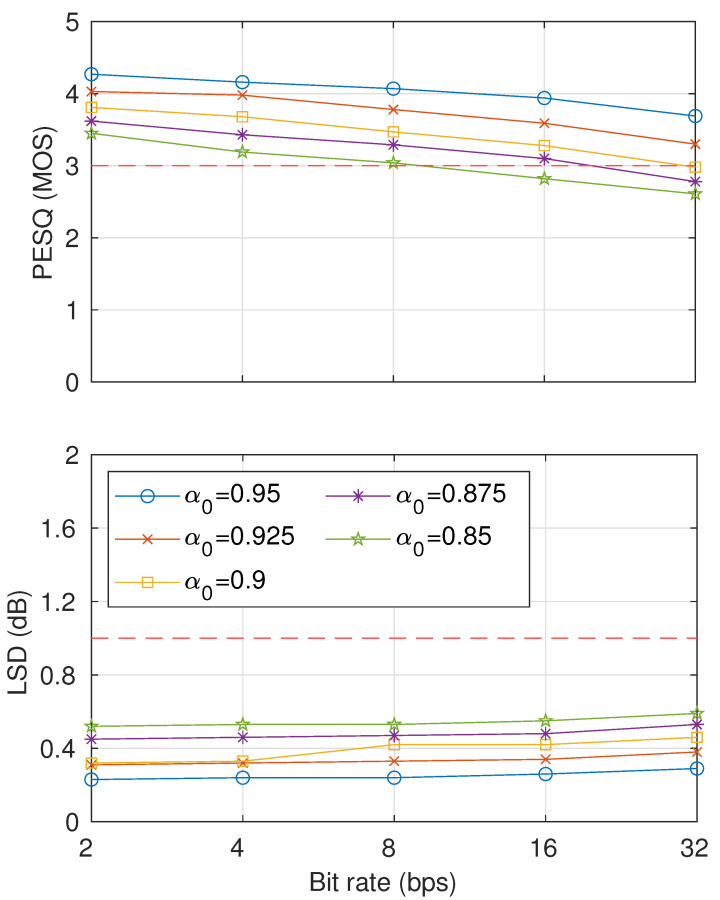
Sound-quality results using several McAdams coefficients for representing bit-“0” ($\alpha_{0}=\left\{0.95,0.925,0.9,0.875,0.85\right\}$) in terms of PESQ (**top**) and LSD (**bottom**).

**Figure 11 entropy-23-01246-f011:**
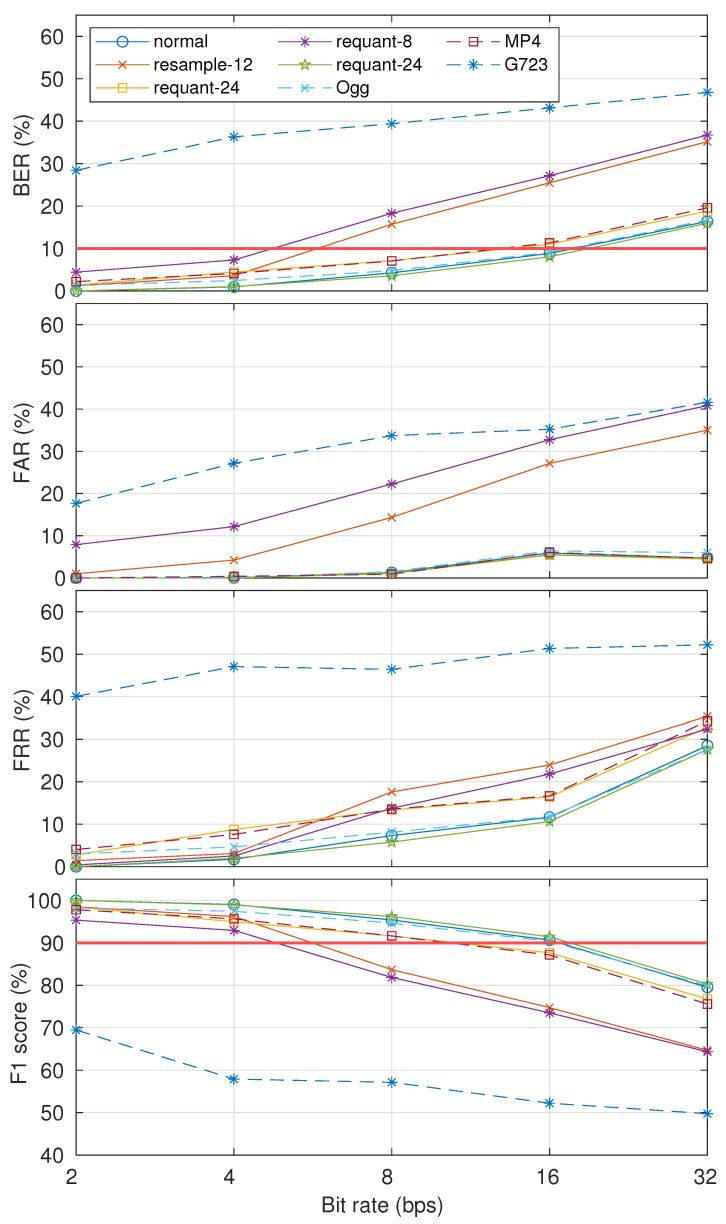
Robustness results in terms of BER, FAR, FRR, and F1-score in eight cases: normal, resample-12, resample-24, requant-8, requant-24, Ogg, G723, and MP4. The McAdams coefficient for representing bit “0” was 0.9 ($\alpha_{0}=0.9$).

**Figure 12 entropy-23-01246-f012:**
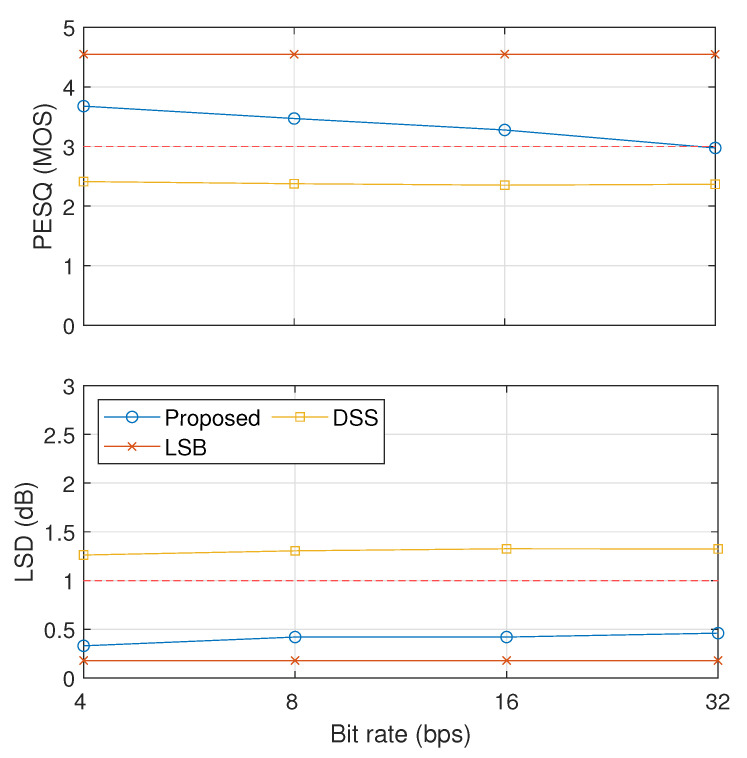
Evaluation of inaudibility results of three compared methods (proposed, LSB, and DSS).

**Figure 13 entropy-23-01246-f013:**
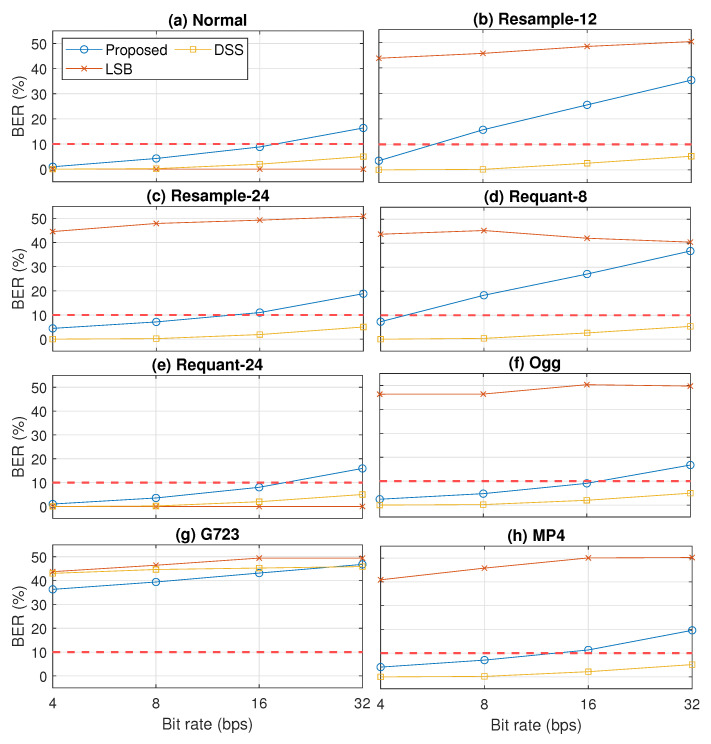
Robustness results of three compared methods (proposed, LSB, and DSS) in terms of BER in eight cases: (**a**) normal, (**b**) resample-12, (**c**) resample-24, (**d**) requant-8, (**e**) requant-24, (**f**) Ogg, (**g**) G723, and (**h**) MP4.

**Figure 14 entropy-23-01246-f014:**
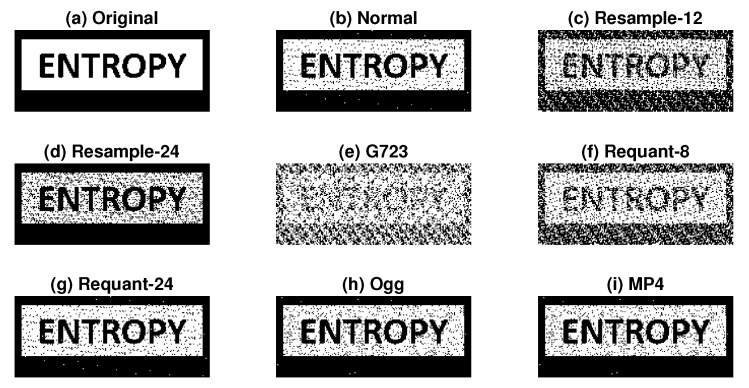
Application of embedding image information using proposed method with 4-bps payload after several non-malicious signal processing operations, i.e., (**a**) original watermark, (**b**) normal, (**c**) resample-12, (**d**) resample-24, (**e**) G723, (**f**) requant-8, (**g**) requant-24, (**h**) Ogg, and (**i**) MP4. McAdams coefficients for representing bit “0” and bit “1” were 0.9 and 1, respectively ($\left(\alpha_{0},\alpha_{1}\right)=\left(0.9,1.0\right)$).

**Table 1 entropy-23-01246-t001:** Statistics of dataset.

Subset	Number of Speakers			Number of Utterances
	Male	Female	Total	
LibriSpeech (train)	4	4	8	225
LibriSpeech (test)	2	2	4	25
VCTK (train)	1	1	2	225
VCTK (test)	1	1	2	25
Total	8	8	8	500

## Data Availability

Not applicable.

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
