# Peer review of "Speech Watermarking Method Using McAdams Coefficient Based on Random Forest Learning"

_entropy, 2021, doi:10.3390/e23101246_

Round 1

Reviewer 1 Report

This paper proposes a speech watermarking method with both acceptable audibility and robustness. 

Concerns: 
-- Are there any strong reasons that random forest is chosen as your classifier?  It is unclear whether the BER/FAR/FRR observed is mainly caused by the difficulty of the watermark detection or because of the limited capacity of random forest. Please include more results of other classifiers.
-- Another concern is that sampling 250 utterances from Librispeech and VCTK seem to be inadequate for detection training. The lack of training data would probably make the trained model hard to generalize to test data from other corpora. Please show the results of random forest on other corpora unseen during training.

Reviewer 2 Report

The authors present their work in speech watermarking by using the formant-shifting based McAdams coefficient method. The manuscript is properly written, seems technically sound, and experiments seem well designed, with sound results. However, some questions raise from reading the text, which are detailed in the remainder section-based review of the article. Please answer the questions in them, in particular the one about the choice for the experimental protocol, which is the one that could be considered most critical if not well rebutted. 

3. Proposed method
- Please ellaborate the differences between the method proposed here and that of [15] "Improving Security in McAdams Coefficient-Based Speaker Anonymization by Watermarking Method"
- Please ellaborate on the choice of these parameters for the McAdams coefficient value
- Please extend the explanation of sections 3.2 and 3.3, which is where the contribution of the manuscript apparently lies. 
- Did the authors consider other classifiers other than random forest classifiers? Please ellaborate a bit

4. Experimental setup
- Did the authors consider balancing the gender of their random sentences? 
- "The total 500 utterances are then split into 90% of
184 the training set and 10% of the testing set" -> Could there not be a problem here if the splits did not preserve disjoint sets in terms of speaker identities? Please clarify if it this separation was considered, and if it was not, please either consider making the corresponding adjustments or explaining why it is not necessary.
- line 194: 'To reach the first goal, we consider five different McAdams coefficients (α = {0.95,0.925,0.9,0.875,0.85}) as representation of bit-“0”, where we keep α = 1 as rep- resentation of bit-“1”.' -> This is a bit confusing, as the authors previously stated their alpha values would be 0.95, 0.9, 0.85. Please clarify why the choice of values is different here.

5. Results
- It is very appreciated that the authors included samples online of their results. In addition, will source code be made available to reproduce performance?

6. Conclusions
- 'Since the McAdams coefficient could be utilized for speaker anonymization, we will consider improving the security of speaker anonymization by using the watermarking approach as our future direction' -> This is a bit confusing, since, as mentioned above, there is a previous work fo the authors in [15] which seems to be doing what is suggested as future work already. Clarification on this point would be appreciated.

stylistic / typos:
line 46: 'the possibility of robustness performance' -> 'the possibility of robust performance', 
line 79: ossilation -> oscillation
line 159: 'we generate two modified speech' -> could this be rephrased? 'two modified speech frames'? or do the authors mean something else?
line 164: 'the watermarked signal (y(n)) is pass' -> 'is passed' / 'passes'

Round 2

Reviewer 1 Report

The revision has answered the questions I raised. 

Reviewer 2 Report

This reviewer appreciates the efforts made by the authors in re-shaping / correcting the article and improving its presentation, making it better suitable for acceptance.